# Improving Milk Yield, Milk Quality, and Follicular Functionality Behavior in Dairy Cows from the Implementation of Microencapsulated Chili Pepper Supplements in Their Diets [note 1]

**DOI:** 10.3390/ani14162361

**Published:** 2024-08-15

**Authors:** Mónica Madrigal-Valverde, Marcus Vinicius Galvão Loiola, José E. de Freitas Júnior, Murilo R. Santiago, Lara Lôbo Dantas, Artur Azevedo Menezes, Isabella de Matos Brandão Carneiro, Gleice Mendes Xavier, Endrigo Adonis Braga Araujo, Juliana Reolon Pereira, Rodrigo Freitas Bittencourt

**Affiliations:** 1Escuela de Agronomía, Campus Tecnológico Local San Carlos, Instituto Tecnológico de Costa Rica, San Carlos, Alajuela 223-21001, Costa Rica; 2Área Académica del Doctorado en Ciencias Naturales para el Desarrollo, Campus Tecnológico Local San Carlos, Instituto Tecnológico de Costa Rica, San Carlos, Alajuela 223-21001, Costa Rica; 3Doctorado en Ciencia Naturales de para el Desarrollo (DOCINADE), Instituto Tecnológico de Costa Rica, Universidad Nacional, Universidad Estatal a Distancia, Costa Rica; 4Escola de Medicina Veterinária e Zootecnia, Campus Ondina, Universidade Federal da Bahia, Salvador 40170-110, Brazil; marcusvinicius@ufba.br (M.V.G.L.); jose.esler@ufba.br (J.E.d.F.J.); murilozootec28@gmail.com (M.R.S.); laralobodantas@gmail.com (L.L.D.); arturmenezes@ufba.br (A.A.M.); isabella.brandao.c@hotmail.com (I.d.M.B.C.); gleicemxavier@gmail.com (G.M.X.); endrigoadonis.vet@gmail.com (E.A.B.A.); rfb@ufba.br (R.F.B.); 5NutriQuest, Campinas 13025-320, Brazil; juliana.pereira@roullier.com

**Keywords:** animal nutrition, animal reproduction, corpus luteum, follicle, tropical dairy, ultrasound

## Abstract

**Simple Summary:**

Microencapsulated hot chili pepper (MHCP) has properties that affect the metabolism of bovines, so the objective of this research was to study the effects of MHCP on milk production and the reproduction of dairy cows. Twenty-four animals were separated into two groups. In the first group, each cow consumed 1 g of microencapsulated hot chili per day. The second group served as a control group for 42 days. Daily milk production and weekly milk composition were monitored for 42 days. These animals were subjected to a hormonal synchronization protocol, tests to monitor the size and vascularization of the ovarian structures, as well as the evaluation of serum progesterone. Ultimately, the supplemented group presented differences in milk production and milk components. On the other hand, a trend was found in which the ovarian follicle was mostly irrigated in the group that consumed the MHCP. We conclude that MHCP provides benefits in dairy production.

**Abstract:**

The present study evaluates the effect of including microencapsulated hot chili pepper (MHCP) in the diet of crossbred dairy cows on the volume and quality of milk and on ovarian morphofunctionality. Twenty-four crossbred females in their lactating period were used. The cows were divided into two experimental groups, a control (CT) and an MHCP -supplemented group (CP) given 1 g a day per animal of microencapsulated hot chili in concentrate for 42 days. Over seven weeks of daily milk production was measured, and sample milk was collected weekly for composition analysis. Animals were subject to an ovulation synchronization protocol on day 0 (D0), and an intravaginal progesterone (P4) implant, estradiol benzoate, and prostaglandin (PGF2α) were administered. On D8, the P4 implant was removed and PGF2α, equine chorionic gonadotropin, and estradiol cypionate were administered to the animals. The ovarian dynamics were evaluated in B mode and color Doppler. There were significant differences (*p* < 0.05) in the group X time interaction, the volume of milk produced, and the amount in kg/day of milk components. There was a higher percentage of vascularization in the preovulatory follicle in the CP group (*p* ≥ 0.10). The findings show that the inclusion of MHCP in the diet of dairy cows does influence their milk production and reproduction.

## 1. Introduction

The different morphotypes of hot pepper contain different levels of vitamin C, total polyphenols, carotenoids, and flavonoids [1,2,3]. In addition, hot chili contains capsaicinoids such as capsaicin. These capsaicinoids are metabolites responsible for the pungent taste in hot chili, and, in turn, they are nutraceuticals to which positive effects on the body are attributed due to their antioxidant, antimicrobial, cardiovascular, antifungal, antiviral, and analgesic effects [2,4]. The above is due to the identification of biochemical compounds, such as the α-glucosidase inhibitor and antioxidant FRAP [5].

In animal health, capsaicin has reported benefits such as cardiovascular protection at a histological level and reduction of triglyceride and cholesterol levels [6]. In turn, the suppression of fat accumulation is associated with the reduction of hepatic triglycerides and intestinal hydroperoxides, which, consequently, have antioxidant, anticarcinogenic, and anti-inflammatory effects [7].

Furthermore, capsaicin has thermoregulatory properties due to its alkaloid content, which include the metabolism of glucose and lipids in ruminants [8]. Modification in lipid metabolism involves an increase in serum insulin, with decreased β-hydroxybutyrate concentrations indicating a decrease in lipolysis in the case of dairy cows supplemented with rumen-protected capsicum [9]. On the other hand, animals exposed to high environmental temperatures reported lower rectal temperatures in the group supplemented with capsaicin [8,10].

Additionally, in ruminants, increased nutrient digestibility and milk quality characteristics such as protein level and, yield of fat-corrected milk and milk solids have been reported in animals supplemented with encapsulated pepper fed at 0.75 or 1.5 g per day [8] and 20 mg/kg [10], including the modification of fatty acids in milk [11] and a reported increase in milk production [9,12].

At the rumen level, modifications in the microbiota have been reported, with increasing populations of *Bacteroides*, *Prevotella*, *Roseburia*, and *Butyrivibrio* in cattle that consumed capsaicin [11].

However, it has been proven that there is a tendency for animals supplemented with capsaicin to increase the amount of dry matter they ingest (*p* = 0.08) [13]. Furthermore, the increase in dry matter consumption has been confirmed when capsaicin is associated with cinnamaldehyde and eugenol [14] and with capsaicin compounds in dairy cows [15].

Finally, in terms of reproductive parameters, the use of 20 to 60 milligrams of capsaicin per kilogram of concentrate per day has indicated positive results, with animals supplemented with the capsaicin compound showing estrus response, ovulatory follicle size at ovulation, and pregnancy rates higher than those of the control group [12].

Therefore, the present study aims to determine the effect of hot chili pepper additive products on ovarian follicular and luteal variables, milk productivity, and performance in dairy cows in a tropical production system.

## 2. Materials and Methods

### 2.1. Animal Care

All procedures carried out in the present study were submitted to and approved by the Committee for Ethics in the Use of Animals (CEUA, abbreviation in Portuguese) of the Federal University of Bahia, protocol number 29/2022.

### 2.2. Location

The experimental work was developed at Entre Rios Experimental Farm (FEER), a property of the School of Veterinary Medicine and Zootechnics of the Federal University of Bahia (EMVZ-UFBA). The forage coverage on the farm corresponds to *Braquiaria decumbens*, with an animal density of 1.2 AU/Ha. The experiment was carried out from September to November 2022.

### 2.3. Population

A total of 24 crossbred females (*Bos taurus* × *Bos indicus*) were used in the lactating period with an open interval of 53.54 ± 11.8 days, an age of 7.16 ± 3.41 years, an average body weight of 447.80 ± 89.6 kg, and an average daily milk production of 6.34 ± 1.91 Liter/cow/day. The experiment was developed with lactating animals in the transition period to a new pregnancy.

### 2.4. Synchronization Protocol

The animals participating in the experiment began a hormonal synchronization protocol on a random day of the estrous cycle called day zero of the experiment (D0). On day zero (D0), a first-use intravaginal device containing 1.0 g of progesterone (São Paulo, Brazil, DIB, Zoetis) was administered, combined with 2 mg of estradiol benzoate intramuscularly (im) (São Paulo, Brazil, Gonadiol, Zoetis) and 12.5 mg of tromethamine dinoprost (im) (São Paulo, Brazil, Lutalyse, Zoetis). On day eight (D8), the intravaginal device was removed and 12.5 mg of tromethamine dinoprost (im) was administered (São Paulo, Brazil, Lutalyse, Zoetis). Simultaneously, 0.6 mg of estradiol cypionate (im) (São Paulo, Brazil, ECP, Zoetisl) and 300 IU of eCG (im) (São Paulo, Brazil, Novormon, Zoetis) were administered.

### 2.5. Ultrasonographic Methods

Preliminarily, the ovarian status of each female to be included in the study was monitored and the corresponding ovarian map was drawn following a gynecological examination. Evaluations in B mode and color Doppler Flow Power were carried out by means of ultrasonographic equipment (Sonoscape S2, Shenzhen, China) using a transrectal transducer with a frequency of 7.5 MHz every 12 h between D8 of the protocol and the moment of ovulation; 96 h thereafter, the intravaginal device was removed. Subsequently, the size and vascularization of the corpus luteum were evaluated on day 11 post ovulation.

### 2.6. Follicular and Luteal Parameters

To assess the ovarian follicular dynamics, images of the two largest follicles were saved to determine the largest follicle on day 8 (FD8) and day 10 (FD10) of the ovulation synchronization treatment regimen every 12 h. To calculate follicular and corpus luteum diameter, follicular and corpus luteum area of the follicle wall, follicular and corpus luteum blood perfusion in the follicle wall, and follicular and corpus luteum blood perfusion in the area of the follicle wall were determined using Brito et al.’s, 2020 methodology.

The follicular growth rate from D8 to D10 (FGR) was determined by the difference between values on FOLA and FOLAD8, which were divided by the growth period and expressed as mm/day. Time of ovulation was determined when the dominant ovarian follicle was no longer detected after being present at the previous ultrasonic examinations (HOV).

### 2.7. Milk Quantity and Quality Parameters

The animals were milked mechanically once a day (6:00 a.m.). The quantity of milk was measured with an MM6 automatic counter (DeLaval, Tumba, Sweden). Effect on yield of 3.5% fat-corrected milk (FCM) was determined according to the equation proposed by Sklan et al. [16], where FCM = (0.432 + 0.165 × fat content in milk) × milk produced (kg/day). A sample of milk was taken from each animal on a weekly basis (0, +7, +14, +21, +28, +35, +42), placed in a polyethylene bottle with 2-bromo-2-nitropropano-1,3-diol (Goiás, Brazil, Bronopol^®^) for preservation, and kept in a cooler with ice at 4 °C. Once all the samples were obtained, they were transferred to the laboratory for lactose quantification, urea nitrogen, protein, somatic cells, fat percentage, total dry extract (TDE), and non-fat dry extract (NFDE).

### 2.8. Blood Collection and Hormonal Dosage

Blood samples were collected on D42 (11 days after ovulation) via the coccygeal vein using 10 mL test tubes without anticoagulants (Vacutainer^®^, Becton Dickinson, Franklin Lakes, NJ, USA). These were immediately placed in a cool box that was maintained at approximately 4 °C. The samples were subsequently centrifuged at 3000× *g* for 15 min for serum separation and then transferred to polypropylene microtubes (1.5 mL) and stored at 20 °C for the subsequent P4 analysis. The progesterone concentration analysis was performed using the chemiluminescence methodology with the fully automated chemiluminescence immunoassay progesterone kit (Maglumi PRG CLIA^®^, Shenzhen New Industries, Shenzhen, China). The test has a measurement range of 2.04–33.58 ng/mL, and the coefficients of intra- and inter-assay variation were 5.57% and 8.82%, respectively. The standard curve range was 0.7–24.2 ng/mL, and the assay sensitivity was 0.13 ng/mL.

### 2.9. Experimental Rations and Food Analysis

The design used in the present study was an unrestricted randomized one where, for supplementation, the animals were divided into two groups at random. The control group (CT, n = 12) received balanced feed with silage, while the treatment group (CP, n = 12) received balanced feed and silage with 1 g/cow/day of the product (Capsin^®^, NutriQuest, Mason City, IA, USA) microencapsulated chili pepper whose main component is capsaicin (trans-8-methyl-N-vanillyl-6-noneneamide). The diets were formulated based on the recommendations of the NRC [17] and were offered once a day (6:00 a.m.) in collective pens with a total size of 250 m^2^; each animal was fed in a long feeder of 73 cm per animal. The animals had access to water ad libitum in a trough with a volume of 572 m^3^.

The duration of the experiment was a total of 42 days, starting after day 40 postpartum. Once a day, consumption/group/day was estimated by quantifying leftovers and collecting them with a broom and shovel. The food remains were placed in a plastic bag (previously weighed) and weighed on a scale.

For the bromatological analysis, the samples of forage, ingredients, and the reject material were dried in a forced air convection oven at 55 °C for 72 h. Subsequently, the samples were ground in a knife mill and passed through a 1 mm sieve for the analysis of dry matter, ash, crude protein, ethereal extract, acid detergent fiber, and lignin. The analyses were carried out under AOAC International standards [18]. The samples were also analyzed for neutral detergent fiber according to Mertens’ methods [19], using thermostable alpha-amylase and sodium sulfite. Non-fibrous carbohydrates were calculated according to the equation proposed by Hall [20]. As for the total digestible nutrients (TND), they were calculated according to the NRC [17], and net lactation energy was calculated as described by Weiss et al. [21]. The indigestible crude protein values were deducted from the dry matter consumption (see chemical and bromatological composition of ingredients used in diets in Table 1).

### 2.10. Statistical Analysis

The analysis was enhanced using the PROC MIXED procedure of SAS statistical software (Statistical Analysis from Windows 9.4, SAS Institute Inc., Cary, NC, USA) according to the model of repeated measures over time with normality of residuals and homoscedasticity and verified using the PROC UNIVARIATE procedure. The comparison of repeated measures was performed using the SAS MIXED procedure.

The DFs were calculated according to the Satterthwaite method (ddfm = sattterth). Autoregression 1 had a covariance structure with lower values according to the Akaike criterion. All parameters were subjected to a mean comparison test using the LSMEANS command. In the case of parameters that did not meet the normality criterion, non-parametric methods were used, namely, the Mann–Whitney test and the Wilcoxon test.

### 2.11. Confidentiality

The publication of the results of this research was carried out in accordance with the provisions of the technical cooperation agreement between the Federal University of Bahia (UFBA) and NutriQuest company (Sao Paulo, Brazil).

## 3. Results

One of the animals in the control group left the experiment because it presented anorexia, so there was a total of 11 animals.

### 3.1. Milk Yield and Composition

For the milk production parameter, there was a significant difference for the time X treatment interaction (*p* = 0.011) (see Table 2). When comparing the treatment effect per week, the difference in the performance of the groups was significant by week seven (*p* < 0.05) (see Appendix A).

Regarding milk quality, when observing the results of the total content of milk components there were no differences for the time X treatment interaction (*p* = 0.142 to 0.881). However, when observing the amount of the components per kilogram of milk, there were significant interactions for the amount of protein, lactose, total dry extract production (TDE), and non-fat dry extract production (NFDE) (see Table 3). For the four parameters, a significant difference (*p* = 0.002, 0.019, 0.020, and 0.034) was observed between the experimental groups in week seven, indicating the superiority of the group supplemented with microencapsulated hot pepper (CP) (see Figure 1).

### 3.2. Ovarian Morphofunctionality

For ovarian morphofunctionality evaluated by B-mode ultrasound, significant differences (*p* < 0.05) were observed for follicular parameters between groups for follicular diameter parameters on D10 as well as for greater follicle diameter (Table 3). The area of the follicular wall and the area of the largest follicle on D8 and D9 presented significant differences (*p* < 0.05) between groups. The differences in both diameter and follicular area indicate that the CT group had larger follicles.

For both the measurement results for the ovulatory follicle growth rate and time between removal of the progesterone device and ovulation there were no differences (*p* ≥ 0.05). But there were differences in the diameter and area of the preovulatory follicle (*p* < 0.05) (see Table 4).

Follicular characteristics evaluated by color Doppler ultrasound do not indicate significant differences between treatments (*p* ≥ 0.05) after repeated measurement (see Table 5). Follicular characteristics evaluated by color Doppler ultrasound only once indicate a tendency for significant differences between treatments (*p* < 0.10) (see Table 6).

Morphofunctional parameters of the corpus luteum (see Table 7) do not indicate significant differences between treatments (*p* ≥ 0.05).

## 4. Discussion

The average volume of milk produced by the animals, both in the control group and in the group that consumed microencapsulated hot pepper (CP) supplementation, is similar to that reported for crossbred dairy cattle in warm climates (5.4–9 L per day) [22].

In another experiment, Chinese Holstein cows were milked three times a day, and during a supplementation period of 45 days there were significant differences in milk production. These evaluations were conducted every five days and the value reported by the researchers is above our average. The reasons lie in the differences among breeds [12]. However, the study agrees with our experiment since the group supplemented with microencapsulated hot chili increased milk production.

In this experiment, experts evaluated different amounts of microencapsulated hot chili pepper (20, 40, and 60 mg/kg), and they concluded that there were significant differences compared with the control group after two weeks of supplementation [12]. Similarly, this information is reported by An et al.’s [10] work in which a dose of 20 mg/kg DM/day of capsaicin was offered.

Increases in dry matter consumption and consequently an increase in milk volume are characteristic during the postpartum period in dairy animals [23]. The increased milk production matches the increased feed intake in groups of animals supplemented with bioactive compounds [24].

The quality of milk reported in percentages for both groups is similar to that recorded for cattle of the Gir breed (primiparous), both in fat content (3.83%), protein (3.67%), lactose (4.73%), and total solids (13.22%) [25].

The amount of fat and protein produced per day for each milk component is below that indicated for European cows (37.5–38.6 g and 34.6–37.5 g, respectively) [26]. However, the protein values are in the range reported for European cows milked once a day (31.7 g/kg) [27]. The main reason is that there are high genetic correlations between milk components and being milked twice a day (0.81–0.99) [27].

For the same crossbreed, Kozerski et al. [28] indicate that animals supplemented with balanced food have amounts above those reported in this experiment: fat (36 g/kg), protein (31.9 g/kg), and total solids (120.8 g/kg). These differences are the result of the diet offered since the fat and protein in milk are directly influenced by the nutritional quality of the food offered to the animal as well as its quantity. In the experiments of Kozerski et al. [28], the animals were offered a greater amount of food than in our study. Moreover, total solids is a composite parameter that includes the amount of protein, fat, lactose, and mineral content of the milk. So, when the protein and fat parameters of the milk are modified, the total solids parameter is also modified.

The components that presented significant interactions (*p* < 0.05) between time and treatment presented a significant difference between the groups (*p* < 0.05) for week seven. In animals supplemented with capsaicin, an increase in FCM is reported, which is in agreement with Vittorazi Jr. et al. [8], where fat, protein, and lactose were is higher compared to the control group (*p* ≤ 0.037).

The variation in protein and lactose is consistent with the variation in the TDE and NFDE parameters. Therefore, the increase of protein and lactose causes an increase in TDE and NFDE. Additionally, milk yield was strongly positively and genetically correlated with both PY and LY (≥0.81) in dairy cattle populations [29].

In the group that received supplementation (CP), the amount of lactose per kilogram of milk produced is close to the parameter reported in Bos taurus grazing cattle, both for purebred cattle (44.1) and for crossbred cattle (45.4) [26]. The CP group maintained a minimum of 40 g kilogram of lactose per kilogram of milk throughout the experiment.

On the other hand, the time variable has shown an effect on the parameters similar to the study by Sauls-Hiesterman et al. [30] in which the volume of milk (kg/day), the composition of milk in terms of fat (%, kg/day), protein (%, kg/day), lactose (%, kg/day), total solids (%, kg/day), and urea nitrogen (mg/dL), changes with time in the control group and in the group supplemented with yeast.

The same behavior is reported when offering diets with different levels of protein to dairy cows in the transition period [31]. Giving capsaicin to dairy animals reveals an increase (*p* < 0.001) in the percentage of lactose in the supplemented group versus the control group [9].

These modifications at the level of milk composition in dairy animals supplemented with capsaicin have been reported for animals affected by heat stress since capsaicin increases protein utilization and thereby supports rumen efficiency. Consequently, this impacts the levels of protein and urea nitrogen secreted in milk [12].

The thermoregulatory action of capsaicin supports the increase in blood circulation that occurs in animals exposed to high temperatures. The increase in irrigated surface for heat dissipation reaches the mammary gland, which consequently affects the adsorption of nutrients and milk synthesis [8].

Modifications in the quantity and quality of milk can be explained due to the reported increase in dry matter consumption in animals supplemented with capsaicin present in bioactive compounds [32,33,34,35]. Furthermore, an increase in water consumption is reported in experimental groups that consumed capsaicin [15,36]. There is a strong correlation between water consumption and DMI (R = 0.98) [33], which allows us to hypothesize its influence on the passage rate.

The above is explained due to the action of spicy aromatic capsaicin compounds on the nervous system (the *vagus* nerve) [36]. In turn, supplementation in ruminants with mixtures of bioactive compounds including capsaicin show modification in the production of the volatile fatty acids propionate, valerate, and isovalerate [15,34,37], as well as an increase in the total amount of VFA [33] and decreased lactate production [15].

The increase in the amount of propionate consequently affects the synthesis of glucose, which reaches the udder and causes an increase in the amount of lactose in milk and, subsequently, the amount of milk.

Additionally, there is an effect on the microbial population of the rumen in animals supplemented with capsaicin, such as the *Butyrivibrio fibrisolvens* bacterium [38], which is an important bacterial population that allows the use of pectin and cellulose derived from plant material [39].

In animals supplemented with capsaicin, an increase in the amount of total solids in milk was reported [12]. On the other hand, the STNG parameter includes the solids of milk other than fat, which explains why a modification in the amount of lactose modifies this parameter.

The milk urea nitrogen (MUN) indicated for crossbred cattle (Gir × Holstein) was (16.17 mg/dL) [28]. These levels are higher than those reported in this study because (MUN) is affected by the amount of protein contained in the diet. In the experiments cited above, the protein reported in the diets and the amount of food offered to the animals was higher than what was offered in this study.

Similarly to the study by Rodríguez-Prado et al. [33] in which dairy animals were supplemented with 250, 500, and 1000 mg/kg of microencapsulated hot chili, the animals in this study that were supplemented with microencapsulated hot chili had a significant decrease in the parameter (linear and quadratic) of urea nitrogen in milk.

Furthermore, somatic cells in milk (SCC), which indicate the immune response of the animal at the time of sampling, show an increase in the migration of defense cells, causing an increase in the SCC. This parameter is commonly used for the detection of subclinical mastitis in dairy animals due to the inflammatory response in the mammary gland [40].

The average somatic cell count per group is below that reported in purebred (175) and crossbred (174) European dairy animals [26].

The somatic cell count has been reported to be lower in dairy animals treated with bioactive compounds from 27 herbs. In the case of somatic cell counts in the experimental groups, good udder health is reported during the experiment.

Regarding the evaluation of reproductive efficiency in dairy females supplemented with doses of capsaicin, there was an improvement in reproductive parameters such as estrus response, ovulation rate, and pregnancy percentage [12]. Furthermore the diameter of the preovulatory follicle on D10 (1.46 ± 0.29 CT and 1.13 ± 0.17 CP) coincides with what was found in our study [41].

The diameter of the ovulatory follicle showed values close to those reported by Rodrigues et al. [42] (11.53 ± 2.64, 11.36 ± 3.02, 11.52 ± 2.98, and 12.13 ± 2.51) as well as those indicated by Ferraz et al. [43] for D8 (8.70 ± 2.10, 9.37 ± 1.40, 10.78 ± 2.31, and 11.67 ± 2.55), D10 (8.78 ± 2.04, 10.67 ± 1.65, 13.12 ± 3.52, and 13.37 ± 2.34), and preovulatory (9.48 ± 2.12, 11.8 2 ± 2.64, 13.66 ± 2.58, and 14.17 ± 2.22).

In general the highest parameters were presented by the control group; in both groups the follicular parameters were within those reported for crossbred animals postpartum.

Differences between the groups where the control group has greater follicular parameters can be explained because capsaicin involves a lower number of short-chain fatty acids in the cow body [6], including cholesterol and cholesterol derivatives such as hormones [44]. FSH, LH, and estradiol are hormones for proper follicular development. This explains why the group supplemented with capsaicin had less follicular development, because when the amount of hormone in the blood decreases, the hormones reach the ovary in less quantity, limiting follicle growth.

In unpublished studies from our group, we have found that in the group supplemented with capsaicin, the amount of serum albumin is decreased. Serum albumin is a transporter of cholesterol derivatives, which reinforces the hypothesis that there was a decline in hormone transport in the group supplemented with capsaicin and this consequently affected follicular growth.

On the other hand, when the overall body condition of the groups at the beginning of the experiment and at the end is compared, a difference is observed in the CT group of +0.03 points and in the CP group of −0.15 points. In Wondie Alemu et al.’s [45] study, dairy animals were separated into two body condition loss groups. In this research, it was found that the group that presented a greater loss of body condition reported a lower hepatic expression of IGF1, IGFBP1, and IGFBP3 (*p* < 0.05). This lower expression of IGF system genes in the liver was associated with lower serum concentrations of IGF1 and, consequently, lower ovarian function.

However, a higher percentage of vascularization of preovulatory follicles indicates a greater amount of blood supplying the ovarian structure. This has a relationship with capsaicin as an agonist of the transient receptor potential vanilloid member 1 (TRPV1), which has various effects on cells, such as membrane fluidity and ion flux; ionic flow is involved with muscle relaxation [46,47] since arteries and veins have muscular layers.

In turn, TRPV1 has been associated with vasodilation [48,49]. In this context, as there is a smaller area to be irrigated and an increase in vasodilation, a greater percentage of the vascularized structure is shown.

The area of the corpus luteum found in this study is like that reported by Rodrigues et al. (2018) (1.24 ±0.70, 1.37 ± 0.57, 1.17 ± 0.69, and 0.87 ± 0.35) for animals subjected to different hormonal synchronization protocols. Furthermore, the percentage of vascularization of the corpus luteum presents great variability [50].

The use of ultrasound with color Doppler allows for the creation of images that record the vascularization existing in the follicular wall [51]. The increase in blood flow to the ovaries causes an increase in the rate of follicular growth, which the reduction in vascularization of theca cells causes follicular atresia [52]. With the development of the corpus luteum, the serum concentration of progesterone increases [52]. Furthermore, the overall conception rate was 44.2% (648/1465), and it was influenced by the luteal blood perfusion score [53].

The morphofunctionality of the corpus luteum indicates that there were no significant differences (*p* ≥ 0.05) between parameters for the experimental groups (see Table 6).

There was a single evaluation of the corpus luteum conducted on D10 post ovulation. Evidence shows a vascularized area, similar to that indicated in this study, for animals with a low antral follicle count (0.97 cm^2^) [54].

On the other hand, in bovine females with follicular evaluations on D8, D9, and D10 and an evaluation of the corpus luteum on D12 after ovulation, both the diameter of the follicle of cyclic females and the size of the corpus luteum have similar dimensions according to the present study (1.87 ± 0.60 CT and 1.87± 1.07 CP) [55].

The vascularization scores (I/low-vascularization area < 40% of the CL; II/medium-vascularization >45% to <50%; and III/high-vascularization > 50%) of Santos et al. [56] provide evidence that in both groups the corpus luteum had high vascularization.

Regarding progesterone concentration, the parameters found in this study are similar to those reported for dairy cows between 7 and 10 days post ovulation [54,55,56,57,58], in which circulating concentrations of P4 were reported to be greater (*p* = 0.02) for low milk production cows and medium milk production animals (2.7 ± 0.1 ng/mL and 2.8 ± 0.1 ng/mL, respectively) [58].

## 5. Conclusions

Offering 1 g per cow per day of microencapsulated hot chili to crossbred dairy cows results in an increase in milk volume and milk components such as lactose and protein, which is associated with dry matter consumption and rumen metabolism.

However, the size of ovarian structures is higher in the control group, which is justified in decrease in fat transport in microencapsulated hot chili group p. But vascularization of the preovulatory follicle wall tended to suffer alterations in the supplemented group.

## Figures and Tables

**Figure 1 animals-14-02361-f001:**
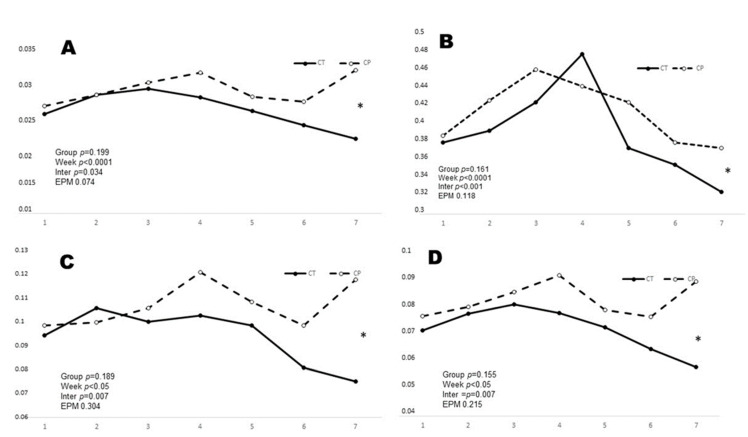
Milk protein (**A**), milk lactose (**B**), total dry extract (TDE) (**C**), and non-fat dry extract (NFDE) (**D**), in the seven weeks postpartum period according to experimental groups (CT and CP). * *p* < 0.05 between groups evaluated. * Significant values between averages within weeks.

**Table 1 animals-14-02361-t001:** Chemical and bromatological composition of the ingredients used in experimental diets based on dry matter.

Item (% of DM)	Supplement	Grass Pangola (*Digitaria decumbens*)	Corn Silage
Ground corn	60.00	-	-
Soybean meal	34.00	-	-
Urea	3.20	-	-
Mineral ^1^	2.80	-	-
Chemical composition (% DM) ^2^			
DM	93.86	88.97	82.59
OM	93.7	91.89	94.97
Ash	6.30	8.11	5.03
CP	25.39	5.60	8.10
EE	4.00	5.50	4.97
NDF	10.95	70.95	47.12
ADF	6.97	39.66	24.81
Total digestible nutrients (g/kg) ^3^	582.46	64.76	73.44
Net energy (Mcal/kg of DM) ^4^	7.46	1.47	1.68

^1^ Contained per kilogram of product = 225 g calcium, 160 g phosphorus, 30 g sulfur, 18 g magnesium, 120 mg cobalt, 2500 mg copper, 120 mg iodine, 1800 mg manganese, 36 mg selenium, 5250 mg zinc, and 1600 mg fluorine. ^2^ DM = dry matter; OM = organic matter, CP = crude protein, EE = ether extract, ^3^ According to NASEM (2021) equation. ^4^ According to NRC (2021) equation. animal unit (standardized to a relative body weight of 450 kg).

**Table 2 animals-14-02361-t002:** Milk yield and composition for the different experimental groups.

Item	Treatment ^1^		EPM	*p*-Value ^2^		
	CT	CP		Group	Week	Inter
Production (kg/day)						
Milk production	8.29	9.59	2.52	0.144	<0.0001	0.011
FCM (3.5%)	8.31	8.91	3.21	0.522	0.010	0.239
Fat	0.03	0.03	0.16	0.780	0.043	0.302
Protein	0.03	0.03	0.07	0.199	0.024	0.034
Lactose	0.04	0.04	0.12	0.161	<0.0001	0.019
TDE	0.10	0.11	0.30	0.189	0.007	0.002
NFDE	0.07	0.08	0.22	0.155	0.007	0.020
Composition						
Fat (%)	3.31	3.15	1.22	0.528	0.195	0.436
Protein (%)	3.11	3.01	0.31	0.217	0.125	0.881
Lactose (%)	4.51	4.50	0.38	0.936	<0.0001	0.725
TDE (%)	11.52	11.39	1.26	0.617	0.129	0.142
NFDE (%)	8.63	8.54	0.64	0.572	0.331	0.418
MUN (mg/dL)	9.64	9.36	3.13	0.638	<0.0001	0.406
SCC (unit/mL)	112.52	154.94	131.47	0.282	0.670	0.462

^1^ Treatment: CT = control (no microencapsulated chili pepper); CP = microencapsulated pepper-supplemented. ^2^ Probability effect for the group, week, and interaction between group and week (Inter). FCM = fat-corrected milk calculated using the equation in Sklan et al. 1992 [16]; TDE = total dry extract; NFDE = non-fat dry extract; MUN = milk urea nitrogen; SCC = somatic cell count.

**Table 3 animals-14-02361-t003:** Follicular characteristics assessed by B-mode ultrasonography of crossbred cows supplemented or not with microencapsulated chili pepper (mean ± standard deviation) with repeated measurement.

Item	Treatment ^1^		EPM	*p*-Value ^2^					
	CT	CP		Group (G)	Week (W)	W × W	W × W × W	G × W	G × W × W
FOLD8 (cm)	0.93 ± 0.10	0.61 ± 0.11	0.113	0.011	0.772	0.669	0.597	0.651	0.683
FOLD9 (cm)	1.21 ± 0.10	0.85 ± 0.11	0.094	0.001					
FOLD10 (cm)	1.40 ± 0.09	1.02 ± 0.09	0.090	<0.001					
FOLD11 (cm)	1.43 ± 0.13	1.06 ± 0.13	0.131	0.001					
FOLAD8 (cm^2^)	0.56 ± 0.08	0.35 ± 0.08	0.083	0.019	0.559	0.931	0.755	0.681	0.745
FOLAD9 (cm^2^)	0.67 ± 0.07	0.45 ± 0.07	0.076	0.009					
FOLAD10 (cm^2^)	0.83 ± 0.09	0.60 ± 0.09	0.109	0.047					
FOLAD11 (cm^2^)	1.05 ± 0.16	0.80 ± 0.15	0.179	0.179					

^1^ Treatment: CT = control (no microencapsulated chili pepper); CP = microencapsulated pepper-supplemented. FOLD8 = diameter of the dominant follicle on day 8; FOLD9 = diameter of the dominant follicle on day 9; FOLD10 = diameter of the dominant follicle on day 10; FOLD11 = diameter of the dominant follicle on day 11; FOLAD8 = total area of the dominant follicle wall on day 8; FOLAD9 = total area of the dominant follicle wall on day 9; FOLAD10 = total area of the dominant follicle wall on day 10; and FOLAD11 = total area of the dominant follicle wall on day 11. ^2^ Probability effect for the group, week, and interaction between group and week: W = linear: W × W = quadratic; W × W × W = cubic.

**Table 4 animals-14-02361-t004:** Follicular characteristics assessed by B-mode ultrasonography of crossbred cows supplemented or not with microencapsulated chili pepper (mean ± standard deviation); single measurement.

Item	Treatment ^1^		*p*-Value ^2^
	CT	CP	
FOLD (cm)	1.44 ± 1.23	1.08 ± 1.12	0.009
FOLA (cm^2^)	1.62 ± 0.25	1.00 ± 0.25	0.025
FOLGR (cm/d)	0.66 ± 0.14	0.50 ± 0.14	0.254
HOV (h)	75.49 ± 5.02	78.66 ± 5.40	0.582

^1^ Treatment: CT = control (no microencapsulated chili pepper); CP = microencapsulated pepper-supplemented. FOLD = diameter of the preovulatory follicle; FOLA = total area of the preovulatory follicle wall; FOLGR = ovulatory follicle growth rate; and HOV = time between removal of the progesterone device and ovulation. ^2^ Probability effect for the group.

**Table 5 animals-14-02361-t005:** Follicular characteristics assessed by color Doppler ultrasonography of crossbred cows supplemented or not with microencapsulated chili pepper (mean ± standard deviation); repeated measurements.

Item	Treatment ^1^		EPM	*p*-Value ^2^					
	CT	CP		Group (G)	Week (W)	W × W	W × W × W	G × W	G × W × W
FOLVD8 (cm^2^)	0.21 ± 0.03	0.17 ± 0.04	0.036	0.276	0.948	0.914	0.898	0.302	0.334
FOLVD9 (cm^2^)	0.32 ± 0.04	0.24 ± 0.04	0.043	0.056					
FOLD10 (cm^2^)	0.41 ± 0.05	0.31 ± 0.04	0.053	0.082					
FOLD11 (cm^2^)	0.45 ± 0.07	0.39 ± 0.06	0.090	0.465					
%FOLVD8	35.58 ± 8.03	39.50 ± 9.12	9.567	0.843	0.307	0.284	0.262	0.246	0.235
%FOLVD9	45.05 ± 8.15	40.17 ± 8.68	8.562	0.576					
%FOLVD10	47.38 ± 7.56	47.51 ± 8.03	8.285	0.987					
%FOLVD11	25.80 ± 11.46	42.77 ± 10.70	14.40	0.254					

^1^ Treatment: CT = control (no microencapsulated chili pepper). CP = microencapsulated pepper-supplemented. FOLVD8 = vascularization area of the dominant follicle wall on day 8; FOLVD9 = on day 9; FOLVD10 = on day 10; FOLVD11 = on day 11; %FOLVD8 = percentage of vascularization in the area of the dominant follicle wall on day 8; %FOLVD9 = on day 9; %FOLVD10 = on day 10; %FOLVD11 = on day 11 of the synchronization protocol. ^2^ Probability effect for the group: W = linear: W × W = quadratic; W × W × W = cubic.

**Table 6 animals-14-02361-t006:** Follicular characteristics assessed by color Doppler ultrasonography of crossbred cows supplemented or not with microencapsulated chili pepper (mean ± standard deviation); single measurement.

Item	Treatment ^1^		*p*-Value ^2^
	CT	CP	
FOLV (cm)	0.41 ± 0.08	0.36 ± 0.08	0.593
%FOLV	37.66 ± 9.66	40.32 ± 9.95	0.079

^1^ Treatment: CT = control (no microencapsulated chili pepper). CP = microencapsulated pepper-supplemented. FOLV = vascularization area of the preovulatory follicle wall; %FOLV = percentage of vascularization in the area of the preovulatory follicle wall. ^2^ Probability effect for the group.

**Table 7 animals-14-02361-t007:** Morphofunctional parameters of the corpus luteum assessed by B-mode ultrasonography and color Doppler, and serum progesterone levels of crossbred cows supplemented or not with microencapsulated chili pepper (mean ± standard deviation).

Item	Treatment ^1^		*p*-Value ^2^
	CT	CP	
CLA cm^2^	3.05 ± 0.35	3.11 ± 0.37	0.900
CLV cm^2^	1.95 ± 0.38	1.95 ± 0.40	0.984
%CLV	67.12 ± 9.68	65.13 ± 10.11	0.858
P4	2.87 ± 1.55	2.20 ± 1.82	0.374

^1^ Treatment: CT = control (no microencapsulated chili pepper). CP = microencapsulated pepper-supplemented. CLA (cm^2^) = area of the corpus luteum on day 11 after ovulation; CLV (cm^2^) = vascularization area of the corpus luteum on day 11 after ovulation; %CLV = percentage of vascularization in the area of the corpus luteum on day 11 after ovulation; P4 = progesterone serum levels on day 11 after ovulation. ^2^ Probability effect for the group.

## Data Availability

The original contributions presented in the study are included in the article. Further inquiries can be directed to the corresponding author.

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
