# Peer review of "Improving Milk Yield, Milk Quality, and Follicular Functionality Behavior in Dairy Cows from the Implementation of Microencapsulated Chili Pepper Supplements in Their Diets [Author-notes fn1-animals-14-02361]"

_animals, 2024, doi:10.3390/ani14162361_

Round 1

Reviewer 1 Report

Comments and Suggestions for Authors

Regarding the article entitled "improving milk yield, milk quality and follicular functionality behaviour in dairy cows from the implementation of chili pepper microencapsulated supplements in their diets **

I have a few questions

1-      What is the relationship between (Chili Pepper) and ovarian follicles?

2-      What criteria were utilized to choose the concentration for the experiment?

3-      The metrics that were measured, such as the amount of milk, are good.

4-     What is your judgment on the economic viability of this material and its application?

5-      Is there an immunological effect? What are your thoughts on various immunological measurements?

6-      Is it possible for positive reproductive results to be the result of using the protocol?

7-      What are the negative effects of this hot chili pepper that might be seen throughout the experiment?

8-      Does it irritate the rumen and intestines?

General Comments

1.      The experiment was good and followed the scientific method

2.      The manuscript is good and scientifically presented

3.      The article is interesting and well written

Comments on the Quality of English Language

Well written English with minor revisions.

Author Response

Comment 1- What is the relationship between (Chili Pepper) and ovarian follicles?

Response 1/ Chili pepper has shown to have effects on vasodilation due to the effect on the muscular layer of the arteries, arterioles and capillaries. By increasing the volume of blood mobilized from the heart to the abdominal aorta, gonadal artery and  the ovarian artery, there is a greater mobilization of nutrients and, potentially, a greater irrigation in the ovarian structures, including the follicles. This process results in a potential improvement in follicular development.

Comment 2- criteria were utilized to choose the concentration for the experiment?

Response 2/ The concentration used in the experiment is based on three foundations: first, the manufacturer's recommendation. Second, the concentration used in similar studies (Abulaiti et al., 2021; 60mg/kg TMR, Animal Production Science; An et al., 2022 40mg/kg/DMI, Animals; Oh et al., 2014, 1000 mg/cow/day Journal of Dairy Science) and finally in our study, the concentrate was formulated according to the requirements of the NRC 2001 for milk production of 12 L/day, considering the estimated pasture consumption of 1.2% of cow weight in NDF. Thus, a concentrate with 25% CP was formulated.

Comment 3 -The metrics that were measured, such as the amount of milk, are good.

Response 3/The amount of milk is correct. Before the experiment, the animals were only fed with grazing, so an expected production was expected for a group raised in a tropical environment. Once the animals in both groups were supplemented, they increased their production of milk due to the nutrients that this product offered. However, results show that the group whose supplementation included the microencapsulated hot chili peppers produced on average 1.3 kg more than the control group.

Comment 4 -What is your judgment on the economic viability of this material and its application?

Response 4/ The economic viability of this material has been analyzed in an ongoing study, in communities where payment is made by volume. There was an increase of 396.5 kg of milk per month in the supplemented group compared to the control group. On the other hand, in companies where the producer is paid for milk quality, a differential income is reported for the production of 20.44 kg of lactose and 10.88 kg of protein per month produced by the supplemented group. This implies a significant economic income for the producer at an expense of $2 per month.

Comment 5 -Is there an immunological effect? What are your thoughts on various immunological measurements?

Response 5/ In some articles, positive results have been observed regarding the immune response of animals. Giallongo et al. (2015) observed that T cell phenotypes were not affected by capsaicin supplementation. Although CAP did not affect numbers of neutrophils positive for phagocytosis, mean fluorescence intensity tended to be quadratically increased (P = 0.08). Supplementation with capsaicin  linearly increased total white blood cells, neutrophils, and eosinophils. The proportion of lymphocytes in total white blood cells decreased linearly and that of neutrophils increased linearly with increasing capsaicin supplementation; therefore, the ratio of neutrophils to lymphocytes was linearly increased by capsaicin. Treatment did not affect concentrations of monocytes and basophils in blood. Red blood cells and hemoglobin concentration quadratically increased and platelets linearly decreased (P = 0.04) with capsaicin, even though mean platelet volume was not affected. Hematocrit percentage tended to be increased by capsaicin. Oxidative stress markers were not affected by capsaicin supplementation.

In this experiment, blood tests were performed on complete blood count, blood biochemistry and oxidative stress. These results have been included in a second article that is currently under review.

Comment 6 -Is it possible for positive reproductive results to be the result of using the protocol?

Response 6/ The use of the hormonal protocol is based on the observation of the ovarian structures in a synchronized manner in all animals, that is, observing the behavior of the follicle and corpus luteum developed over the same number of days for all animals. On the other hand, the potential effects generated by the protocol on the results were controlled using the same protocol with the same management in the group that did not receive microencapsulated hot pepper.

Comment 7 -What are the negative effects of this hot chili pepper that might be seen throughout the experiment?

Response 7/ One of the potential negative effects of using hot chili peppers is that when they are used for a long period of time or are included in the diet of cows throughout their lactation, the TRPV1 receptor is desensitized to the dose of hot peppers used. As a consequence,  the positive effects on the organism are reduced. To avoid this, it is recommended that this additive be used for short periods (less than 60 days), researchers or experts recommend using this additive during periods that cause stress in the animal like the transition period.

Comment 8      Does it irritate the rumen and intestines?

Response 8/The main interest in the use of capsaicin as a natural additive in dairy and beef cattle diets is attributed to its intense effect in stimulating water consumption, food consumption, and the number of meals throughout the day (Calsamiglia et al., 2007; Rodríguez-Prado et al., 2012), which can increase the energy efficiency of the diet in animals on pasture or in confinement.

The feed additive was first incorporated to the mineral mixture and then mixed with the concentrate feeds. The feed additive is a brownish powder with a strong odor containing a minimum of 10 g/kg (minimum) of encapsulated pepper, 5 g/kg of capsaicinoids, besides palm oil and dextrose. According to manufacturer’s information, palm oil and dextrose are intentionally mixed with capsaicin to minimize its pungent taste and possible negative effects in DM intake.

Dextrose and pepper are homogenized and coated with heated palm oil through fluidized bed granulation method to form the encapsulated pepper. Doses were determined based on outcomes reported by Oh studies (Oh et al., 2013, 2015, 2017). A series of experiments with dairy cow conducted by Oh et al. (2013, 2015, 2017) reported positive effects of Capsicum oleoresin on nutrient digestibility, immune response, glucose metabolism, and productivity of dairy cows. In these experiments, however, the feed additive containing Capsicum oleoresin were administered either post-ruminally (abomasum) or top-dressed onto total mixed ration (Oh et al., 2013, 2015, 2017). Note that greater doses than 1500 mg/d of capsaicinoids did not show positive effects or even decreased the performance of dairy cows (Oh et al., 2013). Because of its pungency, Capsicum derived products are usually fed in an encapsulated form to increase acceptability by the animals, and encapsulation may decrease capsaicinoids negative effects on ruminal microbial population and fermentation. Recently, authors reported a tendency for greater DMI and milk protein.

It is important to highlight that the use of Capsicum oleoresin or its mixture with other plant extracts reduces oxidative stress (Karadas et al., 2014; Lee et al., 2003), prevents disease symptoms (Liu et al., 2012; Liu et al., 2013a; Liu et al., 2013b) and improves gut health during normal or disease conditions (Liu et al., 2014) in free ranch poultry and swine.

Reviewer 2 Report

Comments and Suggestions for Authors

Material and method

Because experiment is organized in the same farm, for both experimental and control group, location, and environmental variables, are the same for all the animals, are not a variation sources between experimental groups, and are not necessary to be mentioned.

2.3 population

coefficient of variation for age is 47.6% - age interval is between 4 and 10 years. I speculate the cows are from second to seventh lactation. Form age distribution perspective are experimental groups equilibrated?

The same with body weight and milk production. Some homogeneity tests, before experiment, were performed in this respect?

2.11 On statistical analysis, are not stipulated statistical model followed, and studied variation sources from the model.  

Results.

Data from table 3, is not consistent with data from table 1 in unpublished data. I supposed the experimental groups are inverted.

Regarding the results, significant differences between groups were registered just on milk yield. Differences between milk components, on absolute values are driven by incorporated milk yield effect.  

Discussions

Reference on supplementation of dairy cows with other bioactive compound (polyphenols) is not relevant for the present paper, because were not included in experiment for comparison.

Comparison with diets supplemented with protein out bounded scope of this paper.

In discussions is stated (rows 172-174): “Regarding the evaluation of reproductive efficiency in dairy females supplemented with doses of capsaicin, there was an improvement in reproductive parameters such as estrus response, ovulation rate, and pregnancy percentage”

Estrus response: any activity parameters, or number of jumps were quantified on experimental groups?

Pregnancy percentage was compared between experimental groups?

Also, open period (interval between calving and conception), is irrelevant, because animals enter in synchronization program, main effect of the program being open period reduction.

Regarding negative energetic balance and follicular dimension, milk production is low and differences between experimental groups are not so notable. In fact, such a milk production can be sustained (from the energy point of view) with high quality hay and a proper amount of silage. No combined feed (grains) is requested. In this regard, maybe is better to research the side effect of CP - pepper micro encapsuled supplemented - on the follicular activity.  On the other hand, literature cite an optimum follicular size, (between 12 and 13 millimeters)  correlated with high fertility.

Author Response

Comment 1

Material and method

  • Because experiment is organized in the same farm, for both experimental and control group, location, and environmental variables, are the same for all the animals, are not a variation sources between experimental groups, and are not necessary to be mentioned.

Response 1/ Agreed, we will eliminate environmental variables and modify location characteristics

Comment 2

-population

coefficient of variation for age is 47.6% - age interval is between 4 and 10 years. I speculate the cows are from second to seventh lactation. Form age distribution perspective are experimental groups equilibrated?

Response 2/ Yes, the experimental groups are equalized. We use Diman software for the distribution of the groups. The criteria for equalizing the groups included the age of the animals, body weight, milk production, among others.

Comment 3 - The same with body weight and milk production. Some homogeneity tests, before experiment, were performed in this respect?

Response 3/ The animals were selected and evaluated according to the data from the previous lactation. Furthermore, before each analysis, the homogeneity of variances and normal distribution was analyzed.

A p-value lower than the significance level (usually 0.05) indicated sufficient evidence to reject the null hypothesis, suggesting that the populations were not homogeneous. On the other hand, a p-value greater than 0.05 suggested insufficient evidence to state that the proportions were different, indicating homogeneity between the groups analyzed.

Comment 4 - On statistical analysis, are not stipulated statistical model followed, and studied variation sources from the model.

Response 4/ Yes, the model was not included. However, this mixed model included the mean common to all observations (µ), the effect of the experimental group, the block, the week (time), existing correlations, interactions between the factors and the experimental error.

The following were initially equalized as sources of variation: days postpartum, body condition, body weight, age, milk production. Subsequently, to control the source of variation, part order, blocks were created.

Comment 5 - Data from table 3, is not consistent with data from table 1 in unpublished data. I supposed the experimental groups are inverted.

Response 5/ Yes, the data was included in Table 1 which was unpublished and corrected.

Comment 6 -Regarding the results, significant differences between groups were registered just on milk yield. Differences between milk components, on absolute values are driven by incorporated milk yield effect.

  Response 6/ Yes, we corrected the table “Milk yield and composition for the different experimental groups.”

Comment 7

  • Discussions

Reference on supplementation of dairy cows with other bioactive compound (polyphenols) is not relevant for the present paper, because were not included in experiment for comparison.

Response 7/ Agreed. We will delete it.

Comment 8-Comparison with diets supplemented with protein out bounded scope of this paper.

Response 8/Agreed. We will delete it

Comment 9 -In discussions is stated (rows 172-174): “Regarding the evaluation of reproductive efficiency in dairy females supplemented with doses of capsaicin, there was an improvement in reproductive parameters such as estrus response, ovulation rate, and pregnancy percentage”

Estrus response: any activity parameters, or number of jumps were quantified on experimental groups?

Response 9/When mentioning "estrus response" we refer to animals that showed symptoms of estrus

Comment 10 - Pregnancy percentage was compared between experimental groups?

Response 10/They compared the percentages of pregnancy between the groups in the article mentioned. However, in our study this was not done due to the sample size.

Comment 11-Also, open period (interval between calving and conception), is irrelevant, because animals enter in synchronization program, main effect of the program being open period reduction.

Response 11/Agreed. We will delete it.

Comment 12- Regarding negative energetic balance and follicular dimension, milk production is low and differences between experimental groups are not so notable. In fact, such a milk production can be sustained (from the energy point of view) with high quality hay and a proper amount of silage. No combined feed (grains) is requested. In this regard, maybe is better to research the side effect of CP - pepper micro encapsuled supplemented - on the follicular activity.  On the other hand, literature cite an optimum follicular size, (between 12 and 13 millimeters)  correlated with high fertility.

Response 12/Agreed. We will delete the topic about negative energy balance . And we will include to the fact that the use of hot pepper reduces the short-chain fatty acids circulating in the blood, including cholesterol derivatives such as hormones. This decrease in the flow of hormones in the group supplemented with hot pepper may be related to the difference in follicular size.

Round 2

Reviewer 2 Report

Comments and Suggestions for Authors

from my point of view, paper seems publishable now.